# Branched Ubiquitination: Detection Methods, Biological Functions and Chemical Synthesis

**DOI:** 10.3390/molecules25215200

**Published:** 2020-11-09

**Authors:** Yane-Shih Wang, Kuen-Phon Wu, Han-Kai Jiang, Prashant Kurkute, Ruey-Hwa Chen

**Affiliations:** 1Institute of Biological Chemistry, Academia Sinica, Taipei 11529, Taiwan; jiang790203@gate.sinica.edu.tw (H.-K.J.); prashantkurkute@gate.sinica.edu.tw (P.K.); 2Institute of Biochemical Sciences, College of Life Science, National Taiwan University, Taipei 10617, Taiwan; 3Chemical Biology and Molecular Biophysics Program, Taiwan International Graduate Program, Academia Sinica, Taipei 11529, Taiwan; 4Department of Chemistry, National Tsing Hua University, Hsinchu 30044, Taiwan; 5Department of Chemistry, National Taiwan University, Taipei 10617, Taiwan

**Keywords:** protein ubiquitination, branched ubiquitination, ubiquitin proteasome system, branched ubiquitin polymer synthesis

## Abstract

Ubiquitination is a versatile posttranslational modification that elicits signaling roles to impact on various cellular processes and disease states. The versatility is a result of the complexity of ubiquitin conjugates, ranging from a single ubiquitin monomer to polymers with different length and linkage types. Recent studies have revealed the abundant existence of branched ubiquitin chains in which one ubiquitin molecule is connected to two or more ubiquitin moieties in the same ubiquitin polymer. Compared to the homotypic ubiquitin chain, the branched chain is recognized or processed differently by readers and erasers of the ubiquitin system, respectively, resulting in a qualitative or quantitative alteration of the functional output. Furthermore, certain types of branched ubiquitination are induced by cellular stresses, implicating their important physiological role in stress adaption. In addition, the current chemical methodologies of solid phase peptide synthesis and expanding genetic code approach have been developed to synthesize different architectures of branched ubiquitin chains. The synthesized branched ubiquitin chains have shown their significance in understanding the topologies and binding partners of the branched chains. Here, we discuss the recent progresses on the detection, functional characterization and synthesis of branched ubiquitin chains as well as the future perspectives of this emerging field.

## 1. Introduction

Protein ubiquitination, which is mediated by the consecutive action of E1 ubiquitin activating enzyme, E2 ubiquitin conjugating enzyme and E3 ubiquitin ligase, is one of the most elaborate and versatile posttranslational modifications. Substrates can be modified by a variety of modes ranging from a single ubiquitin molecule to complex polyubiquitin chain. Since ubiquitin contains seven lysine residues and the N-terminal methionine residue capable of conjugating to another ubiquitin molecule, polyubiquitin chain can adopt highly diverse topologies and can be categorized into homotypic and heterotypic chains (Figure 1A) [1,2]. In the homotypic chains, all building blocks of the chain are connected through the same Lys or Met 1 residue and a total of eight different chain types can be formed. Proteomics analyses demonstrated their coexistence in all cell types analyzed, albeit with different abundance [3,4,5]. These structurally distinct ubiquitin chains are recognized by effector proteins with linkage-specific ubiquitin binding domains (UBDs) to result in diverse functional outcomes. For instance, K48- and K11-linked chains are pivotal signals for proteasomal degradation, whereas K63- and M1-chain modified proteins mainly function in cellular signaling. The functions of remaining homotypic ubiquitin chains are less well understood but recent studies have begun to reveal the fates of these atypical conjugates [2,6]. In addition to the homotypic chain, the heterotypic chain, which contains more than one linkage types, can also be formed and can be further categorized into mixed and branched chains. In the mixed chain (also called linear chain), each ubiquitin is modified by only one ubiquitin molecule and connection is mediated through distinct linkages. Conversely, connection of a single ubiquitin to two or more ubiquitin molecules at a time is found in the branched ubiquitin chain.

In theory, a huge number of distinct types of branched ubiquitin chains can be assembled. However, given the technical difficulty in detecting and characterizing branched ubiquitin chains, their existence, abundance and physiological functions remained largely unknown. Not until very recently have several new technologies been developed for the detection and/or quantification of branched ubiquitin chains [7,8,9,10]. These technology advances have not only revealed the unique biological functions of branched ubiquitin chains, but also uncovered their abundant existence [11]. In this review, we will discuss the detection methods and functional outcomes of branched ubiquitination. In addition, since generation of branched ubiquitin chains in vivo often involves complicated processes with the coordinated action of different E2 or E3 enzymes, we will discuss the methods of synthesizing branched ubiquitin chains by chemical or chemical biology approaches.

## 2. Detection of Branched Ubiquitin Chains

To detect the types of ubiquitination including monoubiquitination, multiple mono-ubiquitination and polyubiquitination, one could simply use ubiquitin-specific antibodies to reveal the ubiquitinated patterns in the gel (Figure 1B). This could be coupled with ubiquitin K-R mutants, single-lysine ubiquitin mutants and methylated ubiquitin to reveal whether and which lysine residue is used to form the ubiquitin chains. Alternatively, ubiquitin or its variants labeled with fluorescence probes could be used to reduce the experimental times in detecting the ubiquitination products and to trace ubiquitin chain priming and extension [12]. The conventional method for identifying ubiquitin linkage of the homotypic polyubiquitin chain counts on a linkage-specific antibody to characterize the polyubiquitin chain linkages in vitro or in vivo. On the contrary, detecting linkages of heterotypic polyubiquitin chains often combines several chain-specific antibodies and K-R point mutations to provide indirect evidence for the existence of multiple linkage types. Recently, a K11/K48 bispecific antibody has been developed and used to capture the heterotypic K11 and K48 ubiquitin chains in cell cycle regulation and protein quality control [10]. Although the antibody approach enables the identification of coexistences of two or more than two polyubiquitin linkage types, it cannot distinguish mixed from branched polyubiquitin chains as both chains would react with the antibodies to share the same patterns in the gel-based analysis. An elegant solution is to introduce a tobacco etch virus protease (TEV)-cleaved sequence and a FLAG-epitope sequence at G53 or E64 of ubiquitin for monitoring the TEV-digested polyubiquitinated products. For a K11/K48 branched ubiquitin chain, the branch point consists of two cleaved ubiquitin fragments, thus resulting in a different migration pattern from unbranched chains in the anti-FLAG Western blot analysis [8]. A R54A mutation was later applied to detect the K48/K63 branched chain by removing the trypsin cleavable R54 residue from ubiquitin. In this way, the two GlyGly (di-Gly) modifications at K48 and K63 are preserved in the same trypsinolyzed peptide (L43-R72) for Mass spectrometry (MS) analysis [13]. A K48/K63 chain assembled by the cooperation of two E3 ligases, TRAF6 and HUWE1 was then successfully characterized. This R54A ubiquitin replacement also demonstrated the high abundance of K48/K63 branched chain in mammalian cells. Notably, the utilization of ubiquitin variants for branched chain detection has been limited to only certain chain types, because one needs to design the suitable mutants that can not only diagnose the branched chains but also preserve the normal functions of ubiquitin, such as conjugation pattern and efficiency. The R54A mutation does not significantly defect the ubiquitin chain elongation when tested with multiple E2s [13] and does not affect cell growth when introduced to yeast [14]. Similarly, ubiquitin with a Flag-TEV peptide insertion to G53 or E64 can still confer the polymer formation on substrates when tested in vitro and in vivo [8]. Nevertheless, the ubiquitin variant strategy is unlikely to apply to the detection of all types of the branched chains.

Recently, a collective library of commercially available deubiquitinases (DUBs) has been used to dissect the ubiquitin chain architecture (Table 1), which is termed Ubiquitin Chain Restriction (UbiCRest) (Figure 1B) [15,16]. The in vitro UbiCRest applies selected chain-specific DUB (eight DUBs in total) to digest a particular ubiquitin chain linkage. Therefore, the remnant ubiquitin linkage is different from the cleaved one. The heterotypic chain information can then be displayed after in parallel UbiCRest reactions by multiple DUBs. Hospenthal et al. [15] applied UbiCRest to confirm the composition of K6/K48 polyubiquitination produced by NleL (Non-Lee-encoded effector ligase), a bacterial E3 ligase. Analysis of UbiCRest samples can be performed by gel electrophoresis and Western blot without a dedicated instrument. The gel-based data conveniently provide useful insights into the ubiquitin chain assembly. The recommended experimental UbiCRest procedures are available [16,17]. Nevertheless, there are some limitations by using UbiCRest for detecting branched ubiquitin chains. The UbiCRest cannot well distinguish branched from mixed ubiquitin chains. Some DUBs used in UbiCRest exhibit preference on more than one linkage types. For instance, OTUD3 cleaves K6- and K11-linked ubiquitin chains [18], implying a potential problem in distinguishing K6/K11 branched chains from K6 or K11 homotypic chains. Finally, certain types of branched chains are more resistant to the linkage-specific DUBs, compared with the homotypic chains [13,19]. All of these would impede the accurate diagnosis of branched ubiquitin chains by UbiCRest.

The remarkable difference between the branched and mixed ubiquitin chains is that the branched ubiquitin chain has at least one ubiquitin modified by two (or more than two) ubiquitin molecules. With such scenario, researchers have developed several methods attempting to characterize the branched points directly. Strieter and co-workers combined in vitro limited proteolysis and MS to present the UbiChEM-MS (ubiquitin chain enrichment middle-down mass spectrometry) method for the discovery of branched ubiquitin points (Figure 1C) [7,20]. Trypsin has been commonly used to digest ubiquitin for characterizing the types of ubiquitin chains [21]. The UbiChEM-MS applies minimal trypsinolysis to cleave C-terminal di-Gly residues in the ubiquitin chain. This results in a series of products termed Ub_1−74_, ^GG^-Ub_1−74_ and ^2xGG^-Ub_1−74_ representing the end caped mono-ubiquitin, non-branched ubiquitin and branched ubiquitin, respectively. The UbiChEM-MS has later been applied at the proteomic scale by combining the K11 chain-specific antibody to reveal that ~3–4% of the total ubiquitin population is K11/K48 branched ubiquitin chains accumulated from mitotic arrest state [22]. Recently, the UbiChEM-MS study has advanced our understanding on the Parkin E3 ligases by the identification of enriched K6/K48 branched ubiquitin chains produced by Parkin [23].

The key data in UbiChEM-MS is the population of Ub_1−74_ with different numbers of di-Gly tail(s) from the minimal trypsin digestion of the polyubiquitin chain. However, precise control of minimal proteolysis including time and enzyme activity needs repetitive practices. A viral protein named Lb protease (Lb ^pro^) plays an identical role to the minimal trypsinolysis. Lb ^pro^ cleaves not only its pro-peptide for auto-activation but also various ISGylated and ubiquitinated proteins in the host cells (Figure 1C) [24]. Unlike other deubiquitinases which remove the entire ubiquitin molecule from modified substrate, Lb ^pro^ cleaves ubiquitin at R74, resulting in Ub_1−74_ and di-Gly. Swatek et al. utilized the advantage of Lb ^pro^ and developed the in vitro Ub-clipping method to decode the ubiquitin chain architecture [9]. A point mutation, L102W, of Lb ^pro^ (termed Lb ^pro^*) was introduced for increasing the specificity in clipping Ub_1−74_ with a 7 to 13-fold improved catalytic efficiency to all ubiquitin chains including branched ubiquitin chains. Although it is reported that the catalysis of K27- or K29-linked di-ubiquitin by Lb ^pro^* is less efficient compared to other di-ubiquitin chains, the digestion can be compensated by loading more Lb ^pro^* in the reaction pool for a longer time to achieve similar results. The Ub-clipping method has demonstrated several examples of branched ubiquitination shown by striking intact masses of the cleaved ubiquitin species. For instance, the in vitro polyubiquitin chain generated by UBE2D3 and TRAF6 presents Ub_1−74_, ^GG^-Ub_1−74_, ^2xGG^-Ub_1−74_, ^3xGG^-Ub_1−74_ and ^4xGG^-Ub_1−74_ in which the four di-Gly modifications significantly indicate the existence of ubiquitin with four branched points. A combination of tandem ubiquitin binding entity (TUBE) and Ub-clipping further depicts the branched ubiquitins from HeLa and HEK293 cells similar to the results obtained by UbiChEM-MS [7].

Both UbiChEM-MS and Ub-clipping methods uncover the branched ubiquitin chains directly and can also reveal the ratio of branched to unbranched linkages. However, the chain linkage types at the branched point are not known. One has to combine conventional antibody-based or MS-based analyses to fully decode the ubiquitin chain architecture. Depicting the branched ubiquitin chains at the whole-cell level is not trivial. Both methods have demonstrated successful cases with quantitative information. A wider panel of ubiquitin binders including chain-specific antibody, TUBE [25], trypsin-resistant TUBE [26], ubiquitin affimers [27] or UBDs [28] incubated with whole cellular lysates could pull down specific ubiquitinated populations for the analysis of branched chains. Such combination has not been comprehensively investigated yet and is of the potential to update our understanding of ubiquitin chain assembly in numerous cellular events.

## 3. The Functional Outcomes of Branched Ubiquitination

With the available toolboxes for analyzing branched ubiquitination, our knowledge on this field has been rapidly evolving. To date, a number of substrates have been identified to undergo branched ubiquitination and E2 or E3 enzymes responsible for assembling branched ubiquitin chains have been uncovered. Notably, the branched chains can be built by a single E3 ligase, the sequential action of two E2-E3 pairs or two E3 enzymes. In the last case, the E3 enzyme that adds more ubiquitin monomers to a pre-assembled ubiquitin polymer is also called E4 enzyme [29]. In a number of cases, the E4 enzyme recognizes and is recruited to a particular chain type to branch out the pre-exiting ubiquitin polymer by adding another chain type [13,23,30,31]. At the functional aspect, recent studies collectively indicate that branched ubiquitination endows the substrates a unique signal, which is qualitatively or quantitatively distinct from the homotypic ubiquitination [11,32] (Figure 2). This uniqueness is mediated by effector proteins carrying UBDs and DUBs, i.e., the readers and erasers in the ubiquitin system. Although ubiquitin binding proteins that specifically recognize branched chains have not been identified, formation of branched ubiquitin chain could increase binding affinity or avidity of effectors to enhance the signaling strength [8,33,34]. Alternatively, branched ubiquitin chains could alter the recruitment of UBD-carrying effectors, by either inhibiting effector binding [35] or allowing new effector binding through the extra linkage type [36], thus leading to alterations in the signaling specificity. Branched ubiquitin chains can also be processed differently from the homotypic chains by DUBs [13,19], thus allowing a change in the signaling duration. Given the unique signaling output, it is conceivable that branched ubiquitination is often induced under particular physiological contexts to contribute to specific cellular processes or responses. However, despite the current advances in our understanding of branched ubiquitination, only a small number of branched ubiquitin chain types have been functionally characterized, all of which belong to bifurcated chains. Nevertheless, MS analyses have identified additional branched ubiquitin chain types, which involve two neighboring Lys residues such as K29/K33 and K6/K11 [37,38]. The powerful Ub-clipping method further discovered higher-ordered branched chains with more than two forks [9]. Future studies are needed to unravel their functions. Below, we describe several branched (heterotypic) chain types with known in vivo functions in details. The enzymes that generate branched (heterotypic) chains, DUBs whose catalysis is influenced by branched (heterotypic) chains, and UBDs that bind branched (heterotypic) chains are summarized in Table 2.

### 3.1. K11/K48 Chain: A High-Priority Proteasomal Degradation Signal in Mitosis and Protein Quality Control

The branched ubiquitination was first discovered by studying the action of APC/C [8], an E3 ligase complex driving protein degradation to control mitotic progression [39]. The function of APC/C is mediated by two cognate E2s, UBE2C and UBE2S, which are responsible for ubiquitin chain initiation and elongation, respectively [40,41,42]. Using an engineered ubiquitin variant containing a TEV cleavage site, it was found that UBE2S branches out the short ubiquitin chain assembled by UBE2C to add multiple blocks of K11 chains [8]. Subsequent study with the newly developed K11/K48 bispecific antibody demonstrated that the branched chains generated by APC/C and its two E2s contain K11/K48 linkages [10]. Furthermore, K11/K48 branched ubiquitination was detected on APC/C substrates such as Nek2A and cyclin A in mitotic cells and the global K11/K48 ubiquitination is increased during mitosis. Formation of such branched chains increases the binding of substrates to the ubiquitin receptor RPN10 on proteasome, thereby enhancing substrate degradation [8]. Interestingly, the function of APC/C is beyond the degradation of cell cycle regulators. In human embryonic stem cells, APC/C is recruited to the transcriptional start site during mitosis via WDR5 and TBP, where it assembles K11/K48 ubiquitin chain on histone [43]. The subsequent recruitment of p97/VCP complex and proteasome likely triggers histone degradation to allow chromatin opening and a rapid transcription of pluripotency genes in the next run of cell cycle. These findings explain the function of APC/C in maintaining stem cell properties.

In addition to being decorated on the APC/C substrates during mitosis, K11/K48 branched ubiquitination is induced under various proteotoxic stresses [10]. In mammalian cells, the ubiquitin ligases UBR4 and UBR5 play key roles in the formation of K11/K48 branched ubiquitin chains on misfolded and aggregation-prone proteins such as Huntington’s disease-associated HTT variant, thus allowing an enhanced binding of these modified proteins to p97/VCP complex and proteasomal substrate adaptor HHR23A. Similarly, in the yeast system, the cytoplasmic misfolded proteins are decorated by K11/K48 ubiquitin chain through the combinatory action of two sets of E3 ligases, Doa10 or Hrd1 for generating K11 chain and Ubr1 or San1 for K48 chain [44]. K11/K48 ubiquitin chain is also built on a subset of ER misfolded proteins, known to be degraded by a process called ER-associated degradation (ERAD) [45]. Again, two E3 ligases, UBR4 and KCMF1, are needed to generate such heterotypic chain type in vivo. Notably, in all cases, formation of K11/K48 chain is critical for the clearance of misfolded proteins to prevent their aggregation. Thus, K11/K48 branched ubiquitination plays an important role in protein quality control and its dysregulation may be linked to certain disease states, such as neurodegeneration.

Since K11/K48 branched chain represents an enhanced proteolytic signal, it is conceivable that such ubiquitination is induced under cellular contexts that require a timely degradation, such as the degradation of mitotic regulators during the short M phase and the removal of harmful misfolded proteins under proteotoxic stress. Nevertheless, the topologies of K11/K48 chain built under the two conditions are somewhat different [10]. While APC/C generates multiple blocks of K11 chain, the protein quality control substrates are decorated by multiple blocks of K48 chain. It is currently unknown whether the effectors (i.e., p97/VCP and proteasome) can distinguish these different topologies. Recent structural analyses of branched K11/K48 tri-ubiquitin with X-ray crystallography, NMR spectroscopy, and small-angle neutron scattering have revealed the existence of a novel hydrophobic interface between the two distal ubiquitin molecules. Such branched ubiquitin trimer exhibits a higher affinity in binding proteasome subunit Rpn1 than the homotypic K11 or K48 ubiquitin trimer [34]. However, it is unclear whether the newly identified hydrophobic surface directly involves in Rpn1 binding. Since proteasome contains several ubiquitin receptors with slightly different preferences on binding different ubiquitin chain types, another possibility for an enhanced proteasome binding by the branched ubiquitin chain is the ability to facilitate a coordinated interaction with several ubiquitin receptors [11]. Nevertheless, this hypothesis has not been confirmed experimentally. Thus, there are still several unanswered questions regarding to this first discovered and perhaps the best understood branched ubiquitin chain type.

### 3.2. K63/M1 Chain: A Non-Proteolytic and Combinatory Signal in Regulating NF-κB Pathway

In contrast to the K11/K48 chains, the K63/M1 chains are of a non-proteolytic fate, consistent with the fate of both homotypic M1 and K63 chains. Since literatures related to the K63/M1 chain type do not provide evidence for the formation of branched or mixed chains, we will use “K63/M1 heterotypic chain” throughout this review article. The formation of K63/M1 heterotypic chain is induced upon the activation of various innate immune signaling pathways, including MyD88-, NOD1/2-, TLR3-, and TNFR-dependent pathways [36,46,47]. The generation of such heterotypic chain is mediated by consecutive actions of E3 ligases with different linkage specificities. In response to the pathogens or cytokines, E3 ligases TRAF6 and Pellino 1/2, in conjunction with the E2 enzyme Ubc13-Uev1a, produce K63 chains on the components of innate immune receptor complexes, such as IRAK and RIP. Alternatively, the K63 chains can be unanchored. Both anchored and unanchored K63 chains recruit M1-specific E3 ligase LUBAC, which contains subunits HOIL-1, HOIP and SHARPIN, to assemble M1 chains on K63 chains [36].

Formation of K63/M1 heterotypic chain allows the co-recruitment of downstream kinase complexes through a selective binding to one of these two chain types. In particular, the K63 chain dictates the recruitment of TAK1 complex through the K63 chain-binding ability of its complex component TAB2, as the NZF domain of TAB2 shows a selectivity towards K63 chains [48]. The M1 chain, however, is responsible for the recruitment of IKK complex through the M1 chain-binding ability of UBAN domain in the IKK complex component NEMO [49]. Through such mechanism, the K63/M1 heterotypic chain brings the two kinase complexes to proximity, where TAK1 can activate IKK to allow a rapid NF-κB activation [36,46]. Thus, the K63/M1 heterotypic chain acts as a scaffold to recruit downstream effectors with different linkage specificities. In addition to mediating the assembly of downstream signaling platforms, the K63/M1 chain has an additional role in NF-κB pathway. The K63-chain specific DUB A20 shows a reduced activity for the K63/M1 chain, compared with the homotypic K63 chain [19]. Thus, formation of K63/M1 heterotypic chains could be more resistant to deubiquitination, thereby extending the signaling duration.

### 3.3. K48/K63 Chain: A Versatile Signal for Functional Switch or Output Extension

Similar to the K63/M1 heterotypic chain, the K48/K63 branched ubiquitination has been found to regulate NF-κB pathway. In a mechanism analogous to the generation of K63/M1 chain, the K63 chain assembled by TRAF6 in response to IL-1β stimulation recruits E3 ligase HUWE1 through the K63-chain selectivity of the UBA-UIM domain of HUWE1 [13]. HUWE1 then branches out the preassembled K63 chains to add K48 linkages. Importantly, formation of the K48/K63 branched linkages protect the K63 linkages from CYLD-mediated deubiquitination, thereby extending the duration of NF-κB signaling.

A recent study revealed that K48/K63 branched chain can be an alternative proteolytic signal and this chain type is preferentially associated with proteasome [30]. The proapoptotic protein TXNIP was used as a model to study the mechanism of K48/K63 chain formation and its cellular fate. Similar to the case in NF-κB signaling, the K63 chain is formed initially on TXNIP by E3 ligase ITCH or WWP1 and then recruits K48-specific E3 ligase UBR5 or HUWE1 to assemble the K48/K63 branched chain. Formation of this branched chain is critical for targeting TXNIP for proteasomal degradation, while the K63 homotypic chain is insufficient. Thus, the addition of K48 chain on the preexisting K63 chain facilitates a functional switch to endow the substrate a proteolytic fate.

It is unclear how the K48/K63 branched chain can have context-dependent fates, i.e., proteolytic vs. non-proteolytic. Nevertheless, this difference might be resulted from the existence of other chain type on the substrates. In the NF-κB signaling, a higher-order chain type, i.e., the K63/K48/M1 chain, might be formed, since the K63 chain formed by TRAF6 can recruit E3 ligases HUWE1 (K48-specific) and LUBAC (M1-specific). In line with this idea, HUWE1 knockdown in IL-1β stimulated cells decreases not only the K48/K63 branched chains and the total K63 chains, but also the M1 linkages [13], suggesting the coexistence of K48/K63/M1 linkages. It is possible that the presence of M1 chain alters the fate of K48/K63 chain. With the recently developed Ub-clipping method, it would be possible to determine such higher-order ubiquitin chain type and to elucidate its functional outcome.

### 3.4. K29/K48 Chain: An Emerging Proteolytic Signal

The existence of K29/K48 heterotypic or branched chain in mammalian cells has been detected by pull down analysis with a K29-specific UBD (i.e., the NZF1 domain a DUB called TRABID), followed by a subsequent pull down with a K48 chain-specific UBD or by UbiChEM-MS [7,50]. Since the abundance of this branched chain is elevated upon proteasome inhibition [7], it is thought that this branched chain is also a designated signal for a proteolytic fate. A recent study on yeast ubiquitin fusion degradation (UFD) pathway, which degrades substrates with an N-terminal ubiquitin fusion [51], revealed that UFD substrates are modified by K29/K48 branched chain through a sequential action of UFD E3 enzyme Ufd4p (K29-specific) and E4 enzyme Ufd2p (K48-specific). Importantly, the addition of multiple K48-linked monoubiquitin moieties to the preassembled K29 chain is prerequisite for targeting the UFD substrates for proteasomal degradation [31]. Thus, the K48 chain plays an editing role on the preexisting K29 chain to offer the substrate a degradation fate.

In addition to the sequential action of E3 and E4, K29/K48 branched chain can also be assembled by a single E3. UBE3C, a HECT-family E3 known to synthesize a branched polyubiquitin chain in vitro containing K29 and K48 linkages [52]. Interestingly, this enzyme can function as an E4 for extending the ubiquitin chain on proteasome substrates, thereby promoting their degradation processivity [53]. Recent studies have uncovered an elevated association of UBE3C with proteasome in response to various proteotoxic stresses [54,55] and its role in degrading certain ERAD substrates [45]. It would be interesting to determine whether the K29/K48 branched chain functions as an enhanced proteolytic signal, similar to K11/K48 chain, to enhance protein quality control.

## 4. In Vitro Synthesis of Branched Ubiquitin Chains

Functional study of different ubiquitin chain types from the level of substrate would require the identification of substrate’s chain types. As mentioned above, dissecting the complicated chain types remains technically challenging based on current technologies. Besides the technology advances in detecting branched ubiquitination, the various tools for the in vitro synthesis of branched ubiquitin oligomers not only provide structural insights into the dynamic of branched chains but also open up avenues for understanding the biochemical properties of various DUBs against ubiquitin conjugates branched at specific positions. Thus, in vitro synthesis of branched ubiquitin chains in a sufficient quality via enzymatic or chemical methods would greatly benefit our understanding of the biological meanings of complicated ubiquitin chain types. Below, we discuss the enzymatic, synthetic, and semi-synthetic methods for branched ubiquitin chain.

### 4.1. In Vitro Enzymatic Synthesis of Branched Ubiquitin Chains

The enzymatic process was initially utilized to synthesize homotypic ubiquitin chains. Subsequently, this method has been explored for the synthesis of branched ubiquitin trimers with K48/K63 and K11/K48 linkages. Assembly of the K48/K63 ubiquitin trimer was performed using specific ubiquitin conjugating enzymes E2-25K and Ubc13:Mms2 for K48- and K63-linkages, respectively. To prevent the elongation of ubiquitin chain, ubiquitin variants with K48R/K63R mutations and with an Asp appendage at the C-terminus (ubiquitin-77D) were used as the distal and proximal subunits, respectively [56]. For the assembly K11/K48 branched ubiquitin chain, K11- and K48-linkage specific conjugating enzymes UBE2S and E2-25K were used, respectively, along with the distal ubiquitin with K11R/K48R/K63R mutations and proximal ubiquitin with K63R/77D mutations to control the chain length [34]. For enzymatic synthesis of branched K11/K48/K63 tetra-ubiquitin, E2-25K, Ubc13:Mms2 and UBE2S were used for generating K48-, K63- and K11-linkages and ubiquitin K48R/K63R/K11R variant and C-terminal 6xHis-tagged ubiquitin were used as distal and proximal subunits, respectively [56]. Branched ubiquitin K6/K48 trimer was assembled using a K6- and K-48 specific ligase, i.e., bacterial NIeL ligase, along with ubiquitin K6R/K48R and ubiquitin-ΔG76 [15]. The enzymatic synthesis approach needs the specific E2 and E3 pairs, which impedes this methodology for the assembly of other branched ubiquitin chains with no available enzyme pairs, such as K11/K33 and K27/K29 chains [57]. Furthermore, enzymatic approach assembles the branched ubiquitin chains with different lengths and topologies. These limitations have paved the way for the development of alternative and practical routes in synthetic and semisynthetic methods for generating branched ubiquitin chain with a defined length and topology.

### 4.2. Chemical Synthesis of Branched Ubiquitin Chains

The chemical synthesis method of branched ubiquitin chains was mainly developed with two methodologies, i.e., native chemical ligation (NCL) and hydrazine-based NCL from fragmented ubiquitin peptides that are generated by solid phase peptide synthesis (SPPS). The NCL of peptides derived from SPPS has become a powerful methodology to synthesize various ubiquitin chains. In this methodology, SPPS is used to make medium or smaller peptides which rely on NCL to yield the longer peptides. The NCL reaction involves a trans-thioesterification step in which the nucleophilic sulfur of N-terminal cysteine (Cys) residue attacks the C-terminal thioester bond of another peptide. This intermediate undergoes an S to N acyl shift to afford stable peptide bond. Since the NCL depends on the N-terminal Cys residue, which is generally low in abundance, its application in protein chemistry has been highly restricted. Nevertheless, the desulfurization of Cys yielding alanine has allowed to utilize Cys as a surrogate to alanine, which is one of the most abundant amino acid, thereby empowered the NCL [58]. Various auxiliaries have also been developed to facilitate the desulfurization. The various Lys derivatives such as mercaptolysine with a thiol group at δ carbon of Lys have been developed to mimic N-terminal Cys to construct the isopeptide bond between lysine of substrate and C-terminal thioester of ubiquitin [59,60]. However, the stability issue and difficulties in the synthesis via NCL method have promoted the development of alternative routes to replace thioester [61]. Recently, hydrazine-based NCL has been developed. In this methodology, hydrazine at C-terminus of a peptide reacts with the Cys residue at N-terminus of another peptide to yield a native peptide bond [62]. Branched tri-ubiquitin (K11/K63) was synthesized using the hydrazine-based NCL and acid-labile 1-(2,4-dimethoxyphenyl)-2-mercaptoethyl auxiliary (Figure 3A; Table 3, entry 1). The racemic crystal structure of tri-ubiquitin with D-monomer ubiquitin has shown that Ile44 patches in K11-linked proximal and distal ubiquitin forms an open conformation while in K6-linked ubiquitin forms a close conformation [63]. Alternatively, ubiquitin synthons strategy was used for synthesizing branched K11/48 ubiquitin hexamer. In this strategy, ubiquitin fragments with desired isopeptide linkages were synthesized by SPPS and further elongated with NCL to yield the branched-chain ubiquitin (Figure 3B; Table 3, entry 1) [64].

### 4.3. Semi-Synthesis of Branched Ubiquitin Chains

Although SPPS can generate the native isopeptide linkage, it is limited by the low yield, requirement of skillful and laborious efforts, and the need of refolding. These limitations have led to the development of semisynthetic methods which can be easily performed in any biological laboratory to make the native or mimetic of isopeptide bonds in scalable quantities. Besides the chemical methodologies, semi-synthetic methods have been explored to synthesize isopeptide bond mimetics on a desired lysine position of ubiquitin via genetic mutation of this Lys residue to Cys residue. One type of isopeptide bond mimetics, *i.e.*, thioether linkage, is easy to synthesize and can be digested by DUB. Thiol-ene coupling (TEC) chemistry was used to forge *N^ε^*-Gly-L-homothiaLys isopeptide bond which possesses one carbon atom extra as compared to the native isopeptide bond (Table 3, entry 2) [68]. The proximal recombinant ubiquitin with Cys mutation at a desired lysine position and C-terminal ubiquitin with allylamine were conjugated using radical initiator lithium acyl phosphinate. Ubiquitin trimers with K6/K48, K11/K48 and K48/K63 branched linkages were synthesized using TEC and were subjected to the hydrolysis by various DUBs, including IsoT, A20-OTU and AMSH. The result showed that conjugation at K6 diminishes the A20-OTU activity in K6/K48 trimer. Of note, the inhibition of K48 chain hydrolysis with the presence of K6 chain is in keep with the observation for the decoration of K6 chain on K48 chain-modified Tau protein in Alzheimer disease [3]. Another type of isopeptide bond mimetics is the thioether isopeptide mimetic linkage which substitutes a γ-carbon with a sulfur and is prepared using cysteine aminoethylation-assisted chemical ubiquitination (CAACU) chemistry [79]. In this strategy, auxiliary linker is conjugated to a mutated Cys at a planned position of recombinant ubiquitin to mediate the hydrazine-based NCL with another ubiquitin with C-terminal hydrazine to yield thioether isopeptide bond. C-terminal hydrazine ubiquitin was generated by hydrazinolysis of the ubiquitin 77D using the hydrolase YUH1. Ubiquitin hydrazine on hydrazine-based NCL with auxiliary linked ubiquitin K11C/K48C generates branched K11/K48 ubiquitin chain (Table 3, entry 3) [69]. As previous studies demonstrated that disulfide bond can mimic the isopeptide bond, disulfide-linked K11/K48 branched chain trimer was synthesized (Table 3, entry 4) [70,80]. The pull down assay with 26S proteasome has shown the potential of disulfide bond to mimic the function of isopeptide bond.

### 4.4. Non-Canonical Amino Acids-Assisted Synthesis of Branched Ubiquitin Chains

Expanding the genetic code has been widely researched to synthesize versatile ubiquitin chain linkages that are joined through a native isopeptide linkage or isopeptide bond mimics. This approach requires a bio-orthogonal aminoacyl-tRNA synthetase (aaRS) and its cognate tRNA to direct the incorporation of various non-canonical amino acids (ncAA) into the protein of interest in response to the reassigned stop codon in vivo [81]. The non-enzymatic assembly methodology provides a facile and straightforward platform to generate scalable and homogeneous ubiquitin conjugates with defined chain type and length, which have been the major hurdle for the enzymatic approach. To overcome this hurdle, the elegant genetically encoded orthogonal protection and activated ligation (GOPAL) approach was pioneered to prepare native K6- and K29-linked ubiquitin dimers by making a good use of *Methanosarcina barkeri* (*Mb*) pyrrolysyl-tRNA synthetase (PylRS) tRNA_CUA_ pair, which dictated the incorporation of a ncAA, the *N*^ε^-(*t*-butyloxycarbonyl)-L-lysine (Figure 3C, BocK, **1**) into ubiquitin against a pre-installed amber codon [65]. The remaining amines of the ubiquitin variant were then protected by carboxybenzyl (Cbz) group before the encoded BocK (**1**) was chemoselectively removed under the treatment of trifluoroacetic acid, yielding the acceptor ubiquitin. In parallel, the donor ubiquitin with an essential C-terminal thioester moiety was generated by intein-mediated protein splicing before the accessible amino groups were globally protected by Cbz group. Subsequently, the silver-catalyzed isopeptide chemical ligation was taken place in the presence of the acceptor and donor ubiquitin to generate ubiquitin dimers followed by protecting groups cleavage and protein refolding. The GOPAL-synthesized ubiquitin conjugates possess a native isopeptide linkage as the native ubiquitin chains (Table 3, entry 1). The digestion pattern of various DUBs against the synthesized ubiquitin chains has demonstrated that they behave identically with the native ubiquitin conjugates.

This synthesis method has further been explored to prepare ^15^N-labelled K11/K33 branched ubiquitin trimer for investigating intra-ubiquitin chain interaction by NMR [66]. To prepare the tri-ubiquitin conjugate, BocKs (**1**) were introduced into the proximal ubiquitin at positions 11 and 33 with the aid of *Methanosarcina mazei* (*Mm*) PylRS, and all of the amino groups were protected by allyloxycarbonyl (alloc) group afterward. Different from the GOPAL approach, the distal ubiquitin containing a reactive thioester was prepared by E1 enzyme together with the sodium 2-mercaptoethanesulfonate (MESNA) (Figure 3D). The K11/K33BocK encoded proximal ubiquitin was then reacted with the alloc-protected donor ubiquitin to yield trimeric branched ubiquitin in the presence of silver(I). Finally, complete deprotection of alloc groups of the synthesized ubiquitin chain was carried out before protein renaturation (Figure 3D). Even though the method for branched ubiquitin synthesis using genetically encoded ncAA is limited, there still have been great advances in the synthesis of different types of ubiquitin conjugates. To upgrade the steps in GOPAL approach for avoiding tedious and repetitive protection and deprotection, evolved *Mb*PylRS has been utilized to incorporate protected δ-thiol-L-lysine (Figure 3C, **2**) into ubiquitin at specific K6 position. The lysine analog with a thiol handle can promote NCL in the presence of distal ubiquitin with a C-terminal thioester which was prepared by intein thiolysis. Subsequent desulfurization and protein renaturation allowed the formation of ubiquitin conjugate with a traceless and native isopeptide linkage (Table 3, entry 1) [67]. This approach has also been applied to synthesize ubiquitin-SUMO conjugate for understanding the efficiency and specificity of the C-terminal hydrolase (UCH) family of DUBs [82].

The powerful approach for expeditiously preparing scalable ubiquitin conjugate with a non-hydrolyzable linkage was credited with the click chemistry. The synthesis scheme took advantage of *Mb*PylRS/tRNA_CUA_ pair for inserting *N*^ε^-propargyl-L-lysine (Figure 3C, **3**) into the proximal ubiquitin at the planned position in response to an amber codon, along with the incorporation of azidohomoalanine (Figure 3C, **4**) at C-terminus of the distal ubiquitin by endogenous methionyl-tRNA synthetase in the methionine (Met) auxotrophic *E. coli* [71,72]. The bifunctional ubiquitin variant containing an alkyne and an azide moiety was served as a building block for copper(I)-catalyzed azide-alkyne cycloaddition (CuAAC) that leads to the formation of K11-, K27-, K29-, and K48-linkage specific polyubiquitin chains bearing a 1,2,3-triazole linkage (Table 3, entry 5), respectively. This strategy has also been employed to generate K11-linked polyubiquitin-modified human DNA polymerase beta (Pol β) along with the free K11-linked polyubiquitin chains. The synthesized free K11-linked polyubiquitin chains have further been demonstrated to efficiently suppress the degradation of cyclin B in the *Xenopus* egg extract [71]. In recent years, the ncAA coupled with click chemistry-based ubiquitin chain synthesis method has been utilized to prepare homogeneous ubiquitin chains, including di- [73], poly- [74] and branched ubiquitin conjugates [75]. To achieve the goal, the lysine residues of proximal ubiquitin were mutated to Cys at determined positions before they were further conjugated with additional propargyl acrylate (PA) compound. The addition of PA endowed Cys residues with selective alkyne moieties, in which the functional group is crucially related to CuAAC-based bioconjugation (Table 3, entry 6). Besides, it is indispensable to prepare a C-terminal Aha-encoded distal ubiquitin variant from the Met auxotrophic *E. coli* cells. This strategy was first used to produce seven homogeneous polyubiquitin species by combining PA-modified and Aha-encoded ubiquitin monomer in the presence of copper(I) [74]. The synthesized chains were further subjected to affinity-based proteomic profiling to identify their potential interactors in HEK293T cell lysate. Moreover, the follow-up work demonstrated that this platform is capable of synthesizing the tri-ubiquitin branched at 6/11, 11/48, and 11/63, as well as the tetra-ubiquitin branched at 6/11/48. In all cases, the ubiquitin moieties were linked by triazole linkages (Table 3, entry 6) which are insensitive to DUB digestion [75].

Another approach has exploited the oxime chemistry to synthesize dimeric ubiquitin or ubiquitin-SUMO conjugate which is linked via isopeptide bond mimic with an oxime (Table 3, entry 7) [76]. In the synthesis scheme, the engineered *Mb*PylRS tRNA_CUA_ pair has been used to incorporate *N*^ε^-(tert-butyloxycarbonyl)aminooxylysine (Figure 3C, **5**) and *N*^ε^-photocaged aminooxylysine (Figure 3C, **6**) into ubiquitin at the desired position, respectively. Subsequent deprotection elicits an aminooxy group of the lysine analog to react with the ubiquitin containing a C-terminal aldehyde moiety generated by the intein-fusion technology, thereby producing the DUB-nonhydrolysable, oxime-linked ubiquitin conjugates. In addition, an evolved *Mm*PylRS tRNA_CUA_ pair has been employed to generate Se-alkylselenocysteine (Figure 3C, **7**) [77,83] incorporated ubiquitin, SUMO or substrate proteins. They were further used to synthesize ubiquitin homodimer, ubiquitin-SUMO heterodimers, or site-specific SUMOylated proteins that are covalently coupled via a hydrolysable isopeptide bond mimic (Table 3, entry 8) under native condition. This dehydroalanine-based synthesis has also been achieved using SPPS route through the same chemistry under a denaturing condition [78]. Although the aforementioned two approaches have taken inspiration from different types of chemical ligation, both methods show great potentials for branched ubiquitin synthesis.

Collectively, the ncAA-assisted approaches enable the facile preparation of branched ubiquitin conjugates with required chain types compared to enzymatic method, which needs specific pairs of the E2 and E3 enzymes. Moreover, ncAA-assisted approach is able to make either hydrolysable linkage or non-hydrolysable mimic site-specifically for more intensive biological studies. Nevertheless, the branched ubiquitin conjugates generated by ncAA-assisted GOPAL related or NCL approach still needs to undergo tedious protection, deprotection, and protein refolding which often result in a low yield.

## 5. Conclusions and Perspectives

Over the last few years, the discovery of branched ubiquitin conjugates is considered of vital importance to flourish the complexity of ubiquitin code. With the technology advances in the detection of branched ubiquitination, the biological functions of several branched ubiquitin chain types have been unearthed in vivo [8,13]. The current findings collectively indicate that the branched ubiquitination is recognized or processed differently by the effectors or erasers in the ubiquitin system, respectively, thereby resulting in an alteration in the functional outcomes [19,36]. These functional alterations can be diverse, ranging from qualitative fine tuning to complete switch of the substrate fate [30]. Thus, the emerging evidence has highlighted the importance to dissect the ubiquitin chain type completely to provide better understanding on the cellular decoding mechanism of branched ubiquitination.

It is worth noting that synthetic and semi-synthetic strategies for branched ubiquitin synthesis are arguably dedicated to make homogeneous ubiquitin entities without adding extra mutations as the enzymatic methods, despite the requirement of more elaborate and tedious processes [64,66]. In the future, the synthesized branched ubiquitin conjugates, especially the linkage of which is resistant to enzymatic hydrolysis, might play multifaceted roles in probing unknown interacting partners using pull-down assays. Moreover, the synthesized branched ubiquitin conjugates may be applied to prepare standard fragments for the development of middle-down MS approach. They may also help to accelerate the development of bispecific antibody for cellular branched ubiquitin conjugates detection. Besides, the branched topologies of the synthesized chains may support the design of small-molecule fluorescent probe for detecting different branched ubiquitin types in vitro and in vivo. Undoubtedly, the ultimate goal of the synthesized branched chains is aimed to help the understanding of the underlying mechanism of how they are being assembled in an orderly fashion by enzymes, which will in turn facilitate deciphering the hidden messages of branched ubiquitin code in vivo.

Despite the recent progress on the detection and functional characterization of branched ubiquitination, many outstanding questions remain unanswered in this emerging field. We do not have a complete understanding on the E2 and E3 enzymes that synthesize and DUBs that hydrolyze various branched ubiquitin chains. Moreover, although a number of technologies have been developed to detect the existence of branched ubiquitin chains, none of them are capable of revealing the topology and length of branched chains. Even with the recently developed Ub-ProT method for measuring the total chain length [84], technology breakthrough is still needed to provide insights into the overall architecture of a branched ubiquitin chain, such as the exact arrangement of different linkages. In addition, ubiquitin itself is subjected to several types of posttranslational modifications (PTMs), such as phosphorylation and acetylation [6], but whether and how such PTMs could affect the assembly and disassembly of branched chains remain unexplored. Finally, although a recent study revealed that a large amount of ubiquitin (10–20%) in the total ubiquitin polymers isolated from mammalian cells exist as branched chains [9], only a small number of proteins have been characterized to undergo branched ubiquitination. The current knowledge on the functional outcome of branched ubiquitination is limited to only a few branched chain types. Much effort is needed to identify more proteins modified by branched ubiquitination and to analyze their functional consequences. Since branched ubiquitination can be induced under certain cellular stressed conditions and can function in protein quality control and intracellular signaling implicated in neurodegeneration, aging, and inflammatory and infectious diseases, better understanding of branched ubiquitination would provide new insights into the etiology of these diseases.

## 6. Outstanding Questions

What are the E2, E3, and/or E4 enzymes responsible for the assembly of each type of branched ubiquitin chain? How is the specificity to branched chain formation determined?

How does the formation of branched chains affect the ubiquitin polymer disassembly function of linkage-specific DUBs?

How does the formation of branched chain affect the recognition by effector proteins? Is there effector protein that specifically recognizes the branching point topology?

How could the detailed topology of branched ubiquitin chains be determined?

Whether could each type of posttranslational modification on ubiquitin affect the ability to assemble and disassemble branched ubiquitin chains?

What are the repertoires of substrates being modified by branched ubiquitination? Whether these substrates are responsible for any specialized biological process or function?

What are the combinatorial chemistry methods in synthesizing branched ubiquitin chains with defined topology and PTMs (phosphorylation and acetylation) in native conditions? What are the compatible and practical approaches to install these branched ubiquitin chains on substrate proteins at defined positions?

## Figures and Tables

**Figure 1 molecules-25-05200-f001:**
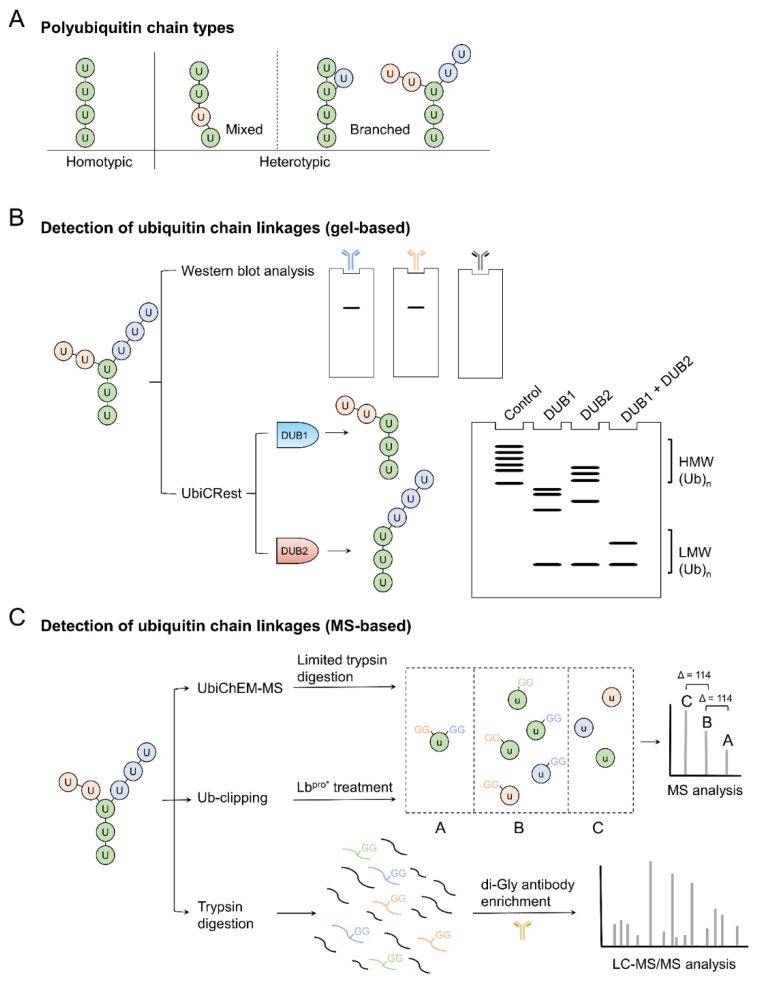
Approaches to detect polyubiquitin chains. (**A**) The polyubiquitin chains include homotypic and heterotypic chains where heterotypic chains are further divided into mixed and branched chains. (**B**) A gel-based approach uses either chain-specific antibody with Western blotting or the UbiCRest deubiquitinase (DUB) library to digest specific chain types. Either approach can detect and provide information of specific chain types from target sample, however, it cannot distinguish branched chain from mixed chains. (**C**) Ubiquitin Chain Enrichment Middle-Down Mass Spectrometry (UbiChEM-MS) and Ub-clipping methods use minimal trypsinolysis or viral Lb protease (Lb ^pro^) to generate Ub_1−74_ with various di-Gly modifications, respectively. The Ub_1−74_ variants (denoted as a small u symbol) could be detected by intact mass (MS) to reveal direct evidence of branched linkages (group A), unbranched linkages (group B), and end-point ubiquitin (group C). The linkage types of branched polyubiquitin chain can be analyzed by LC-MS/MS coupled with trypsinolysis. The combination of two MS-based analyses could offer a better understanding on the topology of a branched polyubiquitin chain.

**Figure 2 molecules-25-05200-f002:**
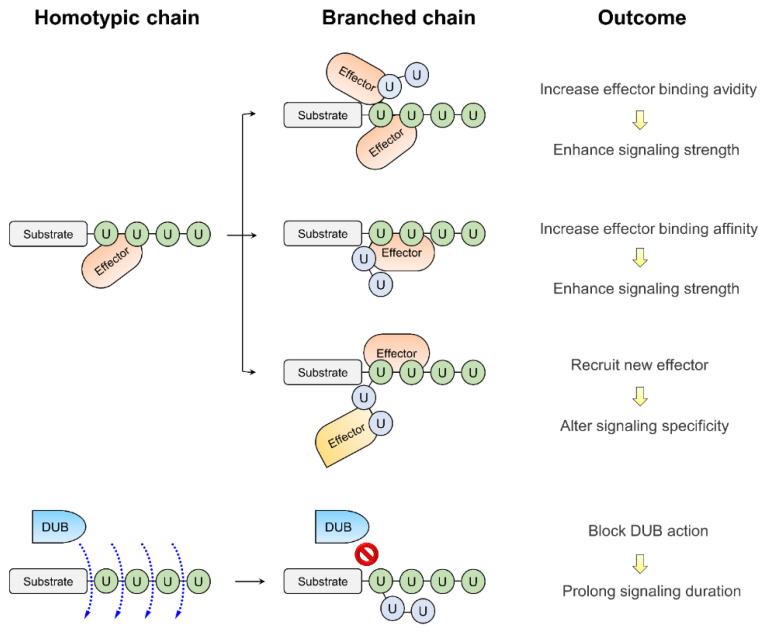
The functional outputs of branched ubiquitination. Formation of branched chains can lead to altered recognition or processing by effectors or DUBs, respectively, thereby changing the functional outputs of modified proteins. See text for details.

**Figure 3 molecules-25-05200-f003:**
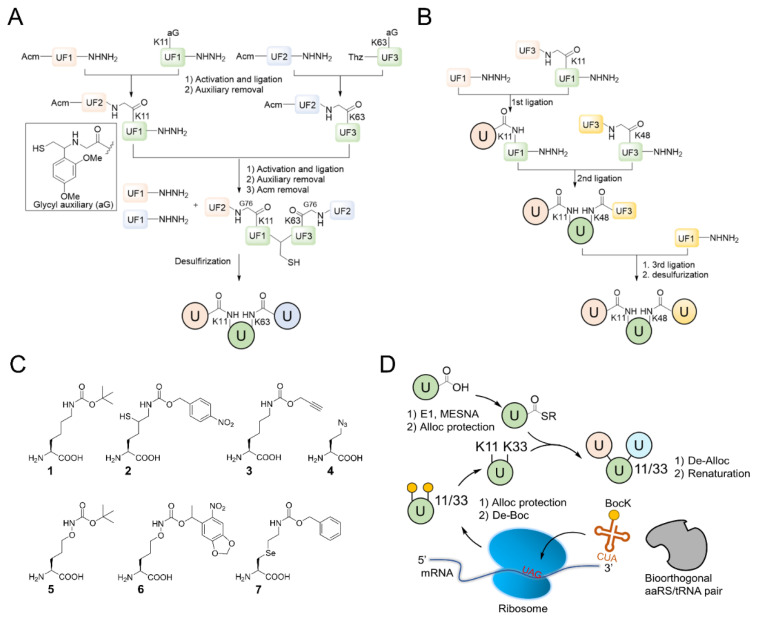
Synthetic and semisynthetic methods of branched Ub synthesis. (**A**) K11/K63 branched tri-ubiquitin synthesis using acid labile auxiliary and hydrazine-based native chemical ligation (NCL). (**B**) Iso-Ub mediated NCL for synthesizing K11/K48 branched ubiquitin trimer. (**C**) Chemical structures of the ncAAs used in ubiquitin chain synthesis. NcAA **1**: *N*^ε^-(*tert*-butoxycarbonyl)-L-lysine (BocK); **2**: (2*S*)-2-amino-5-mercapto-6-((((4-nitrobenzyl)oxy)carbonyl)amino)hexanoic acid; **3**: *N*^ε^-((prop-2-yn-1-yloxy)carbonyl)-L-lysine (Plk); **4**: (*S*)-2-amino-4-azidobutanoic acid (Aha); **5**: (*S*)-2-amino-5-(((tert-butoxycarbonyl)amino)oxy)pentanoic acid; **6**: (*R*)-2-amino-3-((2-(((benzyloxy)carbonyl)amino)ethyl)selanyl)propanoic acid; **7**: (*R*)-2-amino-3-(2-benzyloxycarbonylaminoethylselanyl)propanoic acid (SeCbzK). (**D**) Preparation of K11/K33 branched Ub conjugate using genetic code expansion approach. UF1 = Ub_1–45_; UF2 = Ub_46–75_; UF3 = Ub_46–76_; Acm = *S*-acetaminomethyl; Thz = thiazolidine.

**Table 1 molecules-25-05200-t001:** The commonly used deubiquitinases (DUBs) for UbiCRest [15,16].

DUBs (Favored Ub Linkages)
USP21 (non-specific)
vOTU (non-specific, except M1)
OTUD3 (K6, K11)
Cezanne (K11)
OTUD2 (K11, K27, K29, K33)
TRABID (K29, K33, K63)
OTUB1 (K48)
OTUD1, AMSH (K63)
OTULIN (M1)

Note: USP21, vOTU are used as controls to digest most ubiquitin chains.

**Table 2 molecules-25-05200-t002:** Summary of the enzymes generating branched (heterotypic) chains, DUBs influenced by branched (heterotypic) chains and ubiquitin binding domains (UBDs) interacting with branched (heterotypic) chains.

	Enzyme	Linkage
E2-E3 pairs	UBE2S-APC/C + UBE2C-APC/C	K11/K48 branched
E3s (or E3 + E4)	UBR4 + UBR5	K11/K48 branched
	Doa10 or Hrd1 + Ubr1 or San1	
	UBR4 + KCMF1	
	TRAF6 or Pellino 1/2 + LUBAC	K63/M1 heterotypic
	ITCH or WWP1 + UBR5 or HUWE1	K48/K63 branched
	TRAF6 + HUWE1	
	Ufd4p + Ufd2p	K29/K48 branched
	UBE3C	
DUBs	A20 (inhibited)	K63/M1 heterotypic
	CYLD (inhibited)	K48/K63 branched
UBDs	Rpn10	K11/K48 branched
	Rpn1	
	p97/VCP	
	HHR23A	
	TAK1 (K63), IKK (M1)	K63/M1 heterotypic

**Table 3 molecules-25-05200-t003:** Summary of synthesized isopeptide linkage of Ub conjugates.

Entry	Linkage Structure ^1^	Hydrolysable	Topology	Position	Refs.
1	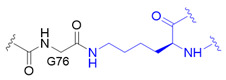	O	homogeneous	6, 29	[63,64,65,66,67]
			branched	11/33, 11/48, 11/63	
2	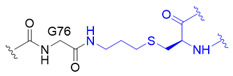	O	homogeneous	48, 63	[68]
			branched	6/48, 11/48, 48/63	
3	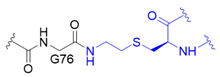	O	homogeneous	6, 27, 33	[69]
			branched	11/48	
4	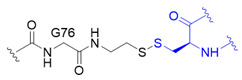	O	branched	11/48	[70]
5	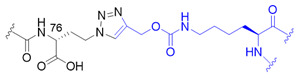	X	homogeneous	11, 27, 29, 48	[71,72]
6	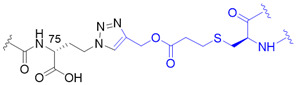	X	homogeneous	6, 11, 27, 29, 33, 48, 63	[73,74,75]
			branched	6/11, 11/48, 11/63, 6/11/48	
7	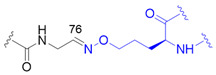	X	homogeneous	6, 48	[76]
8	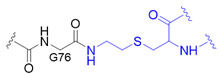	O	homogeneous	11	[77,78]

^1^ Black color represents the corresponding C-terminal G75 or G76 residue of distal ubiquitin; blue color represents the acceptor lysine residue from proximal ubiquitin or ubiquitin-like protein.

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
