# Peer review of "Branched Ubiquitination: Detection Methods, Biological Functions and Chemical Synthesis"

_molecules, 2020, doi:10.3390/molecules25215200_

Round 1
Reviewer 1 Report
The authors provide a timely and thorough review on three aspects of branched ubiquitin chains including detection methods, biological functions and chemical synthesis routes. I enjoy reading this review with a great pleasure. It was well written, easy to follow and included most of up-to-date information on branched ubiquitin chains. Overall I am very enthusiastic to find it to be published in Molecules and believe it fits the broad readership of Molecules.
Some minor concerns are listed below:
- Throughout the manuscript it will be very helpful to include the information that if the work cited is performed in vitro on recombinant proteins, or done in vivo.
- A summary table for all characterized E3 ligases, DUBs and binding proteins to a specific type of branched chain will be informative.
- Are there special binding proteins recognizing the branching point topology?
- Ubiquitin modifications (such as phosphorylation) have been identified and its potential roles in establishment of branched chains can be discussed.
- The mechanism(s) controlling the synthesis of branched chains can also be discussed, eg what controls the switch of E3/E4 to establish the branched chains? Through ubiquitin binding or through post-translational modifications?
Some suggested changes on the manuscript are listed below:
Line 68: a huge number of
Line 73: branched ubiquitin chains
Line 94-104: These ubiquitin mutants are powerful tools for branched chain detection. However, if these mutations affect ubiquitin conjugation pattern/efficiency would need to be discussed.
Line 105-115: a limitation for UbiCRest is the fidelity of DUBs used- and some DUBs demonstrate specificity differences in vitro and in cells.
Line 134: this sentences needs some rephrasing work for clarification.
Line 161: tool boxes
Line 170: brained ubiquitin chains
Line 233: coordinated interaction
Line 242: linkage specificities
Line 237, 256, 258, 264: NF-kB misspelling
Line 265, 282: IL-1b misspelling
Line 280: chain types
Line 295: E4 would needs to be introduced when it is firstly presented.
Line 306: “promote protein quality control”- “enhance protein quality control”
Line 318: K63-linkages
Line 326: 6xHis
Line 395: x-carbon missing
Line 399: the hydrolase YUH1
Line 485: SUMOylated proteins
Line 487: a denaturing condition
Author Response
Throughout the manuscript it will be very helpful to include the information that if the work cited is performed in vitro on recombinant proteins, or done in vivo.
We have indicated the in vitro or in vivo (or in mammalian cells) in all places that were not clear in the original manuscript. In particular, since all proteins mentioned in the “synthesis section” are referred to the in vitro synthesized ones, we simply changed the title as “in vitro synthesis of branched ubiquitin chains and the subtitle as “in vitro enzymatic synthesis of branched ubiquitin chains”.
A summary table for all characterized E3 ligases, DUBs and binding proteins to a specific type of branched chain will be informative.
This is now included as Table 2. In addition, we provided another table (Table 1) to summarize the commonly used enzymes for UbiCRest.
Are there special binding proteins recognizing the branching point topology?
We have searched literatures and cannot find such protein. This is stated in the first paragraph of “the functional outcomes of branched ubiquitination” section and listed as an outstanding question.
Ubiquitin modifications (such as phosphorylation) have been identified and its potential roles in establishment of branched chains can be discussed.
We have discussed this issue in the last paragraph of “Conclusions and perspectives”.
The mechanism(s) controlling the synthesis of branched chains can also be discussed, eg what controls the switch of E3/E4 to establish the branched chains? Through ubiquitin binding or through post-translational modifications?
We have discussed the mechanism of branched chain synthesis in vivo in the first paragraph of “The functional outcomes of branched ubiquitination”.
Some suggested changes on the manuscript are listed below:
Line 68: a huge number of
Line 73: branched ubiquitin chains
Line 161: tool boxes
Line 170: brained ubiquitin chains
Line 233: coordinated interaction
Line 242: linkage specificities
Line 237, 256, 258, 264: NF-kB misspelling
Line 265, 282: IL-1b misspelling
Line 280: chain types
Line 306: “promote protein quality control”- “enhance protein quality control”
Line 318: K63-linkages
Line 326: 6xHis
Line 395: x-carbon missing
Line 399: the hydrolase YUH1
Line 485: SUMOylated proteins
Line 487: a denaturing condition
All of these typos and grammar errors are corrected. However, the “chain type” in Line 280 is referred to a particular branched chain type, that is, the K63/K48/M1 chain. We therefore decide not to change it.
Line 94-104: These ubiquitin mutants are powerful tools for branched chain detection. However, if these mutations affect ubiquitin conjugation pattern/efficiency would need to be discussed.
We agree with the reviewer and discussed the limitations of ubiquitin variant approach at the end of the paragraph.
Line 105-115: a limitation for UbiCRest is the fidelity of DUBs used- and some DUBs demonstrate specificity differences in vitro and in cells.
We discussed a number of limitations for using UbiCRest to characterize branched ubiquitin chains. Nevertheless, UbiCRest is used for in vitro assays, and thus the different specificities between in vitro and in vivo would not be a problem.
Line 134: this sentences needs some rephrasing work for clarification.
The sentence related to the function of Lbpro is rewritten to avoid confusion (on the bottom of p. 7).
Line 295: E4 would needs to be introduced when it is firstly presented.
We introduced the concept of E4 and cited a reference in the first paragraph of “The functional outcomes of branched ubiquitination” section.
Reviewer 2 Report
The authors have in this review given a highly opportune, well written and detailed account of the current state of knowledge of branched ubiquitin chains, detection, functions and experimental synthesis for further exploration. I have provided a short list of minor suggestions of improvements, but I do warmly endorse its publication in the journal.
Abstract section
The authors could include an additional line about branched ubiquitin chain-synthesis and its proposed utility, as it is a large part of the review.
Detection of branched ubiquitin chains section
Line 105-115. It could be stated here (as it is in the figure 1 legend) that UbiCRest cannot distinguish branched from mixed ubiquitin chains.
Section K63/M1 chain: a non-proteolytic …
Line 237-261. Particularly in light of the statement on line 182 (ie “Below, we describe several branched chain types with known functions in detail”) is could be stated that publications referenced in this section do not provide evidence of branched ubiquitin chain types. Also statement Line 263-264, “Similar to K63/M1..” could in this respect be rephrased.
Synthesis of branched ubiquitin chains section
Line 308-313. Although it is detailed in the conclusion section, it could be mentioned already here, briefly, what information and capabilities that can be derived from branched’ chain-synthesis.
Conclusion and perspective section
The section could include a table, box or figure detailing outstanding questions.
The section would benefit from referencing, eg at sentences ending at lines 501, 503, 505 and 514.
Line 533-535. The sentence “In addition, Ub-clipping method revealed…” could be rephrased as it is unclear.
Additional suggestions
Line 20. Replace “resulted” with “is a result”
Line 70. Replace “Only until” with “Not until”
Line 237, 256, 259, 264, 273 and 280. κ is missing in NF- κB
Line 265. 282 β is missing from IL-1β
Line 505. replace “pointed out” with “highlighted”
Author Response
Abstract section
The authors could include an additional line about branched ubiquitin chain-synthesis and its proposed utility, as it is a large part of the review.
We have added two sentences related to ubiquitin chain-synthesis in the Abstract.
Detection of branched ubiquitin chains section
Line 105-115. It could be stated here (as it is in the figure 1 legend) that UbiCRest cannot distinguish branched from mixed ubiquitin chains.
This has been included as one of the limitations for using UbiCRest to characterize the branched ubiquitin chains and presented at the end of UbiCRest paragraph.
Section K63/M1 chain: a non-proteolytic …
Line 237-261. Particularly in light of the statement on line 182 (ie “Below, we describe several branched chain types with known functions in detail”) is could be stated that publications referenced in this section do not provide evidence of branched ubiquitin chain types. Also statement Line 263-264, “Similar to K63/M1..” could in this respect be rephrased.
We appreciate the reviewer for pointing out this. To enhance the accuracy of terminology, we decided to use “heterotypic chains” for describing K63/M1 chains throughout the manuscript. In addition, we stated in the beginning of the K63/M1 chain section that literatures do not provide evidence for the formation of branched or mixed chains.
Synthesis of branched ubiquitin chains section
Line 308-313. Although it is detailed in the conclusion section, it could be mentioned already here, briefly, what information and capabilities that can be derived from branched’ chain-synthesis.
We moved one sentence from the conclusion section to the beginning of this section indicating the capabilities of in vitro synthesized branched chains.
Conclusion and perspective section
The section could include a table, box or figure detailing outstanding questions.
The outstanding questions are listed in the final part of the manuscript and will be presented as a “box” in the article.
The section would benefit from referencing, eg at sentences ending at lines 501, 503, 505 and 514.
All references are cited in the first and second paragraphs of the “Conclusions and perspectives” section.
Line 533-535. The sentence “In addition, Ub-clipping method revealed…” could be rephrased as it is unclear.
We have revised the sentence (in the last paragraph of the “Conclusions and perspectives” section) to improve the clarity.
Additional suggestions
Line 20. Replace “resulted” with “is a result”
Line 70. Replace “Only until” with “Not until”
Line 237, 256, 259, 264, 273 and 280. κ is missing in NF- κB
Line 265. 282 β is missing from IL-1β
Line 505. replace “pointed out” with “highlighted”
All these words are changed in accordance to reviewer’s suggestions.